# Dynamic Linkages among Climate Change, Mechanization and Agricultural Carbon Emissions in Rural China

**DOI:** 10.3390/ijerph192114508

**Published:** 2022-11-04

**Authors:** Taifeng Yang, Xuetao Huang, Yue Wang, Houjian Li, Lili Guo

**Affiliations:** College of Economics, Sichuan Agricultural University, Chengdu 611130, China

**Keywords:** climate change, agricultural carbon emission, agricultural mechanization, agricultural production

## Abstract

Climate change has become a major environmental issue facing all countries, having a significant effect on all aspects of agricultural production, such as the agricultural mechanization process and fertilizer use. Greenhouse gases produced by agricultural machinery and fertilizers during agricultural production are an important cause of climate change. On the basis of the above facts, researching the connection between agricultural mechanization, climate change, and agricultural carbon emissions is crucial for the development of low-carbon agriculture and for addressing climate change. We used a variety of econometric models and methods to analyze data from China’s multiple provinces (cities) covering the years 2000 through 2019, in order to meet the research objectives. Furthermore, we utilized rainfall and sunlight as variables to assess climate change and adopted Granger tests to establish the link between rainfall, sunlight, agricultural mechanization, and carbon emissions in farming. The findings indicate a bidirectional causality relationship between rainfall, sunlight, agricultural mechanization, and carbon emissions in farming. Rainfall and sunlight are Granger causes of agricultural mechanization. Furthermore, agricultural mechanization has favorable effects on carbon emissions of agriculture, and climate change has long-term implications on agricultural mechanization and carbon emissions of agriculture. Finally, this paper investigated the green path suitable for the low-carbon development of Chinese agriculture, arguing that the government should formulate low-carbon agricultural policies by region and actively promote the upgrading of agricultural machinery.

## 1. Introduction

Climate change has become a common problem faced by all countries. During the 20th century, the average global temperature rose about 1 degree Celsius [1], which led to sea level rise [2], species extinction [3], and frequent climate extremes [4]. China’s carbon dioxide emissions hit 11.9 billion tons in 2021, making up 33 percent of the world’s total and far outpacing those of other nations and regions, according to the International Energy Agency (IEA). As a consequence, controlling China’s carbon emissions is vital for mitigating climate change. The Chinese government has also shown its determination to tackle global warming. China has pledged to peak its carbon emissions by 2030 and become carbon neutral by 2060, which is referred to as the “dual carbon” target. In China, about 20 percent of carbon emissions are attributable to agricultural production and land use [5], and agriculture-related carbon emissions make a contribution of about a quarter of the globe’s total [6]. Due to the extensiveness and universality of China’s agricultural production activities and the dispersion of agricultural production entities, coupled with the wide range, randomness, concealment, difficulty in monitoring, and quantification of agricultural carbon emissions, China’s agricultural carbon emissions are difficult to control. Meanwhile, China‘s agricultural production is large and growing rapidly. If there are no corresponding reduction measures, agricultural greenhouse gas emissions will continue to rise rapidly. Therefore, in order to achieve the “double carbon” goal, China must find a way to reduce agricultural carbon emissions as soon as possible, as well as realize the transformation of agricultural production to green, low-carbon, and sustainable development.

Most agricultural GHG (greenhouse gas) emissions currently come from fertilizer, feed, pesticides, and energy consumption, as revealed in the research conducted by World Food and Agriculture Organization [7]. Numerous studies have shown that climate change increases the volatility of agricultural production and has a differential impact on different regions, but climate change damages agricultural production in general [8]. Farmers and the government have also taken corresponding actions to comply with climate variation. Examples include cultivating arable land and increasing the use of chemical fertilizers to cope with agricultural production reduction brought by bad weather [9], increasing the use of pesticides to address pests and diseases caused by temperature rise [10], and investing in more mechanical equipment and changing irrigation methods to deal with extreme droughts and floods [11]. Each country should use agricultural machinery on the basis of climatic conditions [12] and guide the agricultural mechanization following its requirements. Therefore, the direction and pace of agricultural mechanization will be influenced by climate change.

From 1980 to 2015, China’s agricultural mechanization continued to deepen, which enhanced agricultural productivity [13]. However, due to the uncentralized distribution of agricultural land in China, the use efficiency of agricultural machinery was hampered. Giant energy use was caused by the inefficient operation of equipment, which aggravates carbon emissions to a large extent [14]. In the last few years, energy consumption brought about by agricultural mechanization has gradually become a crucial source of carbon emissions in farming production [15]. Therefore, investigating the connection between agricultural carbon emissions, climate change, and agricultural mechanization in the backdrop of “dual carbon” today can make up for the vacancy in research in related fields and provide suggestions on measures to promote the growth of low-carbon agriculture for the government.

Accordingly, China Statistical Yearbook and China Rural Statistical Yearbook covering 2000 to 2019 are the data sources for the analysis of this study. We chose rainfall and sunlight as variables to measure climate change, and then used ARDL (autoregressive distributed lag model) and PVAR (panel data vector auto regression) to look into the connection between climate change, agricultural mechanization, and agricultural carbon emissions.

The key contributions of this paper in comparison to earlier material are listed below. First, climate change and agricultural carbon emissions have been hot issues in recent years, but there are few empirical studies on the interaction between them. At the same time, most of the literature adopts only one primary explanatory factor, while the empirical study in this paper ingeniously integrates multiple variables including agricultural mechanization, climate change, and agricultural carbon emissions, demonstrating the interaction and transmission mechanism between them. Second, the dynamic test framework established in this paper considers the cross-sectional dependence, co-integration relationship, hysteresis effect, and different mutual inspection methods. The purpose is to guarantee the precision and reliability of the research results and provide a reference for the subsequent research of other scholars. Finally, this paper puts forward recommendations for advancing energy-saving and low-carbon agriculture, as well as establishing low-carbon agricultural policies.

## 2. Literature Review

A substantial amount of carbon emissions are now produced by China’s agriculture, drawing the interest of numerous academics. They have discussed the elements that influence carbon emissions in China’s farming sector, believing that human activities such as pesticides, fertilization, irrigation, mechanization, and mulch coverage significantly contribute to carbon emissions in farming production [16]. In recent years, due to extreme droughts and floods caused by climate change, mechanization has become an indispensable factor in agricultural carbon emissions [17]. Mechanized equipment has significantly improved labor productivity and resource utilization, further contributing to the agricultural technological progress and output growth [18]. However, agricultural machinery is mainly driven by fossil fuels, and the gas generated by fossil fuel combustion significantly harms the environment [19]. Unfortunately, there has been little reference to climate change in the discussion of the factors that contribute to agricultural carbon emissions.

Comprehensive research has already been conducted on how climate change would affect the agricultural industry. By and large, climate change has uncertain impacts on agriculture in different regions, but mainly adverse effects [20]. Grain security is seriously threatened by climate variation since agricultural production depends on relatively consistent weather patterns [21]. Changes in rainfall patterns, floods, droughts, temperatures, and pests and diseases have hindered the development of agriculture, adversely affecting crop production and yields [22,23,24]. Additionally, climate change affects the direction and process of agricultural mechanization. The continuous severe weather is unfavorable to the input and use of machinery and equipment [25]. According to studies, climate change in northern Norway has made spring sowing windows even more constrained, affecting soil working days and farmers’ management of agricultural mechanization [26]. Therefore, countries should use agricultural machinery based on climate conditions and guide mechanization following its requirements. Although the existing literature has confirmed the impact of climate change on the agriculture department, few scholars have looked into the connection between climate change and agricultural carbon emissions.

However, many scholars have found that climate change has affected the factors contributing to agricultural carbon emissions. We can roughly divide the factors that lead to the connection between climate change and agricultural carbon emissions into natural and artificial factors. In terms of natural factors, climate change considerably influences the carbon sequestration efficiency of farm production. Although crop cover can slow down the erosion rate of heavy rainfall on soil, the increase in rainfall poses a threat to the extent of soil erosion in farmland [27]. On the one hand, soil erosion can destroy soil’s physical and chemical properties, resulting in the loss of soil nutrients and the decline of carbon sequestration. On the other hand, soil and water loss caused by heavy rainfall leads to the accumulation of sediments carrying large amounts of chemical fertilizers and pesticide residues in rivers and reservoirs, which adversely affects the carbon cycle in the atmosphere [28]. From the perspective of human factors, farmers have taken corresponding agricultural production measures according to the changing climate conditions [29], which may indirectly affect carbon emissions. Soil moisture deficiency caused by reduced rainfall can reduce the biological function of crops, which is more likely to induce pests and diseases [24]. Increased pesticide use can reduce agricultural output losses caused by illnesses and insects, but it can be damaging to the environment [30]. In addition, researchers found that adverse climatic conditions affected soil processing conditions in certain parts of the UK, leading to a corresponding reduction in the use of farm machinery [31]. Moreover, farmers may extract more groundwater for agricultural irrigation to ease the pressure on water use from changing temperature and rainfall patterns. However, groundwater extraction consumes energy and exacerbates agricultural carbon emissions [32]. Farmers may also use drip irrigation, sprinkler irrigation, and other water-saving irrigation techniques in agricultural production to cope with climate change. However, sprinkler irrigation technology requires high water pressure, and the application of irrigation equipment may increase GHG emissions [33]. Therefore, climate change is closely related to agricultural carbon emissions through human adaptation behavior, especially the input of mechanized equipment. Overall, we should not ignore the link between carbon emissions and climate change.

Through the literature, we note that the existing research focuses on the effects of agricultural mechanization on agricultural carbon emissions and the effects of climate change on agricultural mechanization. Although a great deal of research has confirmed the association between climate change and agricultural carbon emissions, few scholars have taken both climate change and mechanization into account when studying the factors that contribute to carbon emissions. Additionally, previous studies mostly used single provincial, municipal or regional data when examining the effects of mechanization or climate change on agricultural departments, but few studies chose data from multiple provinces or regions of a country. Therefore, this paper thoroughly examined the data covering China’s multiple provinces (cities); innovatively took rainfall and sunlight as variables to measure climate change; and incorporated agricultural mechanization, climate change, and agricultural carbon emissions into a homogeneous measurement framework. The purpose was to look into the long-term relationship between agricultural mechanization, climate change, and carbon emissions in agriculture, especially the relationship between the latter two, in order to fill the gap in existing research and propose countermeasures and suggestions for sustainable growth of ecological agriculture.

## 3. Materials and Methods

### 3.1. Calculation of Agricultural Carbon Emission

Agricultural carbon emissions (ACE) chiefly include fossil fuels, deforestation, burning, fertilizer use, arable land, and animal digestion and excretion [34]. This study focused on carbon emissions from agriculture diesel inputs, selecting fertilizers, pesticides, agricultural irrigation, agricultural electricity, agricultural farming, etc. The collected data were then multiplied by the usage of carbon sources and carbon emission factors using the carbon emission factor method to derive the total agricultural carbon emission of each province (municipality). At the same time, in order to avoid the length of the article, the empirical methods and results of this paper used ACE to represent agricultural carbon emissions.
$$E=\sum E_{i}=\sum (T_{i}*\delta_{i})$$
where E represents the total agricultural carbon emissions, $E_{i}$ represents the agricultural carbon emissions of different carbon emission sources, $T_{i}$ represents the carbon source consumption, and $\delta_{i}$ represents the carbon emission factor. Table 1 lists the emission coefficients of each carbon source for the readers’ reference.

### 3.2. Data and Variables

In this paper, agricultural mechanization intensity (total power of agricultural machinery/cultivated land area) was used to measure the degree of agricultural mechanization. Climate change will have many impacts on the Earth’s environment, including rainfall and the duration of sunlight [37], which play an important role in agriculture [38]. The direct effect of these two factors on agricultural carbon emissions is relatively rare in previous studies, and the correlation between the two is small (compared to other climate variables such as temperature). Therefore, this paper chose rainfall and sunlight as variables representing climate change for further empirical study. This study adopted the annual data of 30 provinces (municipalities) from the China Rural Statistical Yearbook (2000–2019) and the China Statistical Yearbook (2000–2019) for empirical analysis. To carry out the research smoothly, this paper set the following variables on the basis of previous literature: agricultural carbon emission (PERAGCARBON), rainfall (RAIN), sunlight (SUNLIGHTS), and agricultural mechanization index (MACHINE). To alleviate heteroscedasticity, we analyzed the data by logarithmic processing. In addition, we provide readers with the definitions of the above variables in Table 2.

### 3.3. Descriptive Statistic

Table 3 shows descriptive statistics for each variable from 2004 to 2019. The average carbon emission was 0.313025, and the interval was 1.055501, which indicates that the data had volatility. At the same time, the average rainfall and sunlight were also different from the interval. Thus, the climate conditions in the 30 provinces (municipalities) were diverse. At the same time, there were differences in agricultural carbon emissions and agricultural mechanization in 30 provinces (cities), which could have a symphony for farmers’ income and planting area. At the same time, the standard deviation of the original data was too large, so we logged the data to reduce the impact of spurious regression and heteroscedasticity The standard deviation after the logarithm of the data was larger, indicating that the data were more dispersed.

### 3.4. Test for Cross-Sectional Dependence

One of the characteristics of panel data is that there is a varying degree of correlation between cross-sections, leading to the correlation between cross-sectional heterogeneity or regression errors, which affects the unbiasedness, consistent, and valid estimation of standard panel data [39]. Therefore, before panel testing, cross-sectional correlation tests should be performed in order to address cross-section correlation [40].

The Breusch–Pagan Lagrange multiplier (LM) test was proposed by Breusch and Pagan in 1980 to test cross-sectional correlation. Pesaran improved the disadvantages of the Breusch–Pagan LM test and proposed the Pesaran cross-sectional dependence (CD) and Pesaran LM tests. Subsequently, some scholars have proposed the bias-corrected LM test and extended it to the dynamic panel data model [41]. We provide the calculation formula of the section test method in Appendix A for the readers’ reference.

### 3.5. Panel Unit Root Tests

When common least squares are used, the presence of horizontal unit roots in panel data can result in pseudo-regression, leading to false estimates. Therefore, the variable unit root tests should be performed first. In addition, the unit root of a non-stationary sequence can be eliminated by difference [42]. On the basis of previous studies, this paper used the Levin–Lin–Chu test (LLC), the Im–Pesaran–Shin test (IPS), Fisher’s augmented Dickey–Fuller (ADF) test, the Breitung t-stat method, and Fisher’s Phillip and Perron (PP) test in order to examine the panel unit root. We provide the specific process of these five methods in Appendix B for the readers’ reference.

### 3.6. Panel Cointegration Test

The paper used the Kao test [43] to conduct a co-integration test on panel data. According to the test results, we can judge if there is a long-term relationship between variables. This study lists two main steps of the test:

Step 1: Set each section to have the same coefficient and different intercept:(1)$$\;y_{it}=\alpha_{i}+\beta_{1}x_{1i}+\beta_{2}x_{2i}+\beta_{m}x_{mi}+e_{it}$$

In the above equation, $\alpha_{i}$ represents a single intercept parameter, and $e_{it}$ represents the residual.
$$e_{it}=\rho_{i}e_{it-1}+v_{it}$$

Step 2: Perform the unit root test on the residual sequence of step 1. The null hypothesis is $H_{0}:\rho =1$, and the following statistics are constructed:(2)$$\;ADF_{t}=\frac{t_{\rho }+\frac{\sqrt{6N}\delta_{v}}{2\delta_{0v}}}{\sqrt{\delta_{0v}^{2}/\left(2\delta_{v}^{2}\right)+3\delta_{v}^{2}/(10\delta_{0v}^{2}})}$$

### 3.7. Granger Causality Test

After the above research, we focused on the causal relationship between the variables. Granger [44] first proposed a method to analyze the causality of time series data. On this basis, Dumitrescu and Hurlin [45] improved it. This paper adopted the improved Granger causality test of Dumitrescu and Herlin. This paper lists the formulas for the causality test:(3)$$y_{i,t}=\alpha_{0,i}+\alpha_{1,i}y_{i,t-1}+\cdots +\alpha_{k,i}y_{i,t-k}+\beta_{1,i}x_{i,t-1}+\cdots +\beta_{k,i}x_{i,t-k}+\varepsilon_{i,t}$$
(4)$$x_{i,t}=\alpha_{0,i}+\alpha_{1,i}x_{i,t-1}+\cdots +\alpha_{k,i}x_{i,t-k}+\beta_{1,i}y_{i,t-1}+\cdots +\beta_{k,i}y_{i,t-k}+\varepsilon_{i,t}$$
where $x_{i,t}$ and $y_{i,t}$ are the observed value of individual *i* in period *t*, respectively, and *k* represents the number of lags of the individual. 

### 3.8. Autoregressive Distributed Lag

To better understand the dynamic relationships between the variables studied, an autoregressive distributed lag model (ARDL) proposed by Pesaran [46] was used. The ARDL model has many advantages, such as dealing with different delay-order variables and analyzing statistical regression and other common models. The typical ARDL models are as follows:(5)$$\;\phi \left(L,P\right)y_{t}=\overset{k}{\underset{i=1}{\sum }}\beta_{i}\left(L,q_{i}\right)x_{it}+\delta w_{t}+u_{t}$$
(6)$$\;\phi \left(L,P\right)=1-\phi_{1}L-\phi_{2}L^{2}-\cdots -\phi_{p}L^{p}$$
(7)$$\beta_{i}\left(L,q_{i}\right)=1-\beta_{i1}L-\beta_{i2}L^{2}-\cdots -\beta_{iq_{i}}L^{qi}$$

In the above equation, *P* represents the order of lag $y_{t}$ and $q_{i}$ represents the order of lag of the *i*-th independent variable $\;x_{it}$. *L* is the lag operator, which can be defined as follows: $Ly_{t}=y_{t-1}$, and $w_{t}$ is the determination vector of S row 1 column.

### 3.9. FMOLS and DOLS

The paper used two static models, FMOLS (full modified ordinary least squares) [47] and DOLS (dynamic ordinary least squares) [48], to ensure the robustness of the results in Section 3.7. Compared with the OLS (orthogonal least square method) model, these two models construct a second-order bias consisting of endogenous bias and non-central bias.

The panel FMOLS estimator *β* is given by
(8)$$\;\beta_{NT}^{*}=N^{-1}\overset{N}{\underset{i=1}{\sum }}{\left[\overset{T}{\underset{i=1}{\sum }}{\left(X_{it}-{\bar{X}}_{i}\right)}^{2}\right]}^{-1}\overset{T}{\underset{i=1}{\sum }}\left(X_{it}-{\bar{X}}_{i}\right)\gamma_{it}^{*}-T_{\hat{\tau_{l}}}$$
(9)$$\gamma_{it}^{*}=\left(\gamma_{it}-{\bar{\gamma }}_{i}\right)-\frac{\hat{L_{21l}}}{\hat{L_{22l}}}\Delta X_{it},\hat{\tau_{l}}=\hat{\Gamma_{21l}}+\hat{\Omega_{21l}^{0}}-\frac{\hat{L_{21l}}}{\hat{L_{22l}}}\left(\hat{\Gamma_{22l}}+\hat{\Omega_{22l}^{0}}\right)$$

The DOLS is written as follows:(10)$$\gamma_{it}=\alpha_{i}+\beta_{i}X_{it}+\overset{jl}{\underset{j=-ji}{\sum }}\theta_{ij}\Delta X_{it-j}+\varepsilon_{it}^{*}$$
where the estimated coefficient *β* is given by
(11)$$\;\beta_{DOLS}^{*}=N^{-1}\overset{N}{\underset{i=1}{\sum }}{\left(\overset{T}{\underset{t=1}{\sum }}Z_{it}Z_{it}^{i}\right)}^{-1}\left(\overset{T}{\underset{t=1}{\sum }}Z_{it}\gamma_{it}^{*}\right)$$
where $Z_{it}=\left(X_{it}-\bar{X_{l}},\Delta X_{it-j},\ldots ,\Delta X_{it+k}\right)$ is 2 (*K* + 1) vectors of regressors.

### 3.10. Impulse Response Approach and Variance Decomposition

To further study the dynamic influence of each variable, this paper adopted impulse response and variance decomposition analysis to grasp the mutual influence and action of each variable dynamically. The impulse response function refers to the impact trajectory brought about by one standard deviation of random disturbance terms on the current and future values of itself and other endogenous variables, which can describe the dynamic influence among variables more intuitively. The variance decomposition can determine the influence of explanatory variables and other variables on endogenous variables by the degree of contribution of each shock to endogenous variables. Lanne [49] proposes that the estimated impulse response exists in the vector autoregressive (VAR) model as follows:(12)$$\;y_{t}=\overset{k}{\underset{j=0}{\sum }}\phi_{i}y_{t-i}+\varepsilon_{t}$$

$\phi_{i}$ in (12) is the impulse response function, which can be transformed into infinite moving regression estimation by the following equation:(13)$$\;\phi_{i}=\left\{\begin{array}{c} I_{k}\;,\;i=0\\ \overset{i}{\underset{\dot{J}=1}{\sum }}\phi_{t-j}A_{j},\;i=1,2,\ldots \end{array}\right.$$

$A_{j}$ is a coefficient matrix that transforms VAR into an infinite vector moving average, k is the optimal lag term, and $\varepsilon_{t}$ represents the error term.
(14)$$y_{it+h}-E\left[y_{it+h}\right]=\overset{h-1}{\underset{i=0}{\sum }}\varepsilon_{i}\left(t+h-1\right)\emptyset_{i}$$
(15)$$\overset{hI}{\underset{i=0}{\sum }}\theta_{nm}^{2}=\overset{hI}{\underset{i=0}{\sum }}{\left({i^{\prime }}_{m}K\emptyset_{in}\right)}^{2}\;$$

Equation (15) represents the contribution of variable *N* to the prediction error variance of variable *M* in period *h*.

## 4. Results

### 4.1. Cross-Sectional Dependence Tests Results

The results of cross-sectional correlation tests in Table 4 and Table 5 show that the null hypothesis of no cross-sectional correlation at the significance level of 1‰ was rejected by all four tests. Therefore, the cross-section of panel data was correlated.

### 4.2. Unit Root Test Results

LNRAIN and LNSUNLIGHT both rejected the null hypothesis at a significance level of 1% in each of the five trials, but the other two variables only rejected the null hypothesis in the LLC trial. Therefore, we performed a first-order differential and tested the data, wherein the final result rejected the zero hypothesis for the unit root of each variable at the critical 1% level. However, this suggests that there may be a spurious regression and therefore we need to use the KAO test for cointegration. We show the results of the five unit root test methods in Appendix C for reference. For details see Table A1 The results of panel unit root tests.

### 4.3. Panel Cointegration Test Results

Table 6 and Table 7 show the co-integration test results of rainfall, agricultural mechanization, and ACE, and the co-integration test results of sunlight, agricultural mechanization, and ACE, respectively. Both groups rejected the original hypothesis, indicating that the panel data existed in a cointegration relationship. The results confirm a long-run equilibrium cause-and-effect relationship between these two sets of variables and help us further study the effects of climate change and agricultural mechanization on ACE.

### 4.4. Results of DOLS and FMOLS

Table 8 and Table 9 show the estimation results of FMOLS and DOLS panel models. From the level of explicitness of parameters, DOLS had a better fitting effect. Thus, we know that when the agricultural mechanization index increases by 1%, ACE will increase by 0.24%. When rainfall increases by 1%, ACE will increase by 0.88%. When sunlight increases by 1%, ACE will decrease by 0.61%, indicating that rainfall and agricultural mechanization have a certain positive correlation with agricultural carbon emissions, while sunlight is negatively correlated with ACE.

### 4.5. Results of ARDL

We used the Akaike information criteria (AIC) to determine the optimal lag length of the model. Finally, the model for agricultural carbon emissions, agricultural mechanization, and rainfall was ARDL (2, 3, 3), while the model for agricultural carbon emissions, agricultural mechanization, and sunlight was ARDL (1, 1, 1).

The long-term relationship of variables obtained from the ARDL model is shown in Table 10. In the long term, agricultural mechanization and rainfall will increase ACE, consistent with the results shown in Section 4.4. In the short term, looking at the first-order difference data, the results showed that agricultural mechanization will promote ACE. However, the impact of rainfall on ACE in the short term is not strong, indicating that the impact of rainfall on ACE may be a long-term process.

The effect of agricultural mechanization on ACE in Table 11 is similar to the conclusion in Table 11. Sunlight can inhibit ACE in the long and short term, but the impact of short-term sunlight on ACE is not obvious, and thus it is speculated that the impact of sunlight on ACE is a long-term process, such as with rainfall.

### 4.6. Granger Causality Tests

Granger causality exists in the previous cointegration, but we cannot guarantee that the causal relationship between variables was identified. Therefore, we further conducted the Granger causality test. Table 12 and Table 13 show the results of the causality test among several groups of variables. The null hypothesis is that there is no Granger causality between variables. We generated Figure 1 to facilitate readers’ understanding of the causal relationship between variables.

We found bidirectional causality between agricultural mechanization and ACE, bidirectional causality between rainfall and ACE, and bidirectional causality between sunlight and ACE at the 5% significance level. In addition, sunlight and rainfall were unidirectional Granger causes of agricultural mechanization at the 5% level of significance.

### 4.7. Impulse Response and Variance Decomposition Results

#### 4.7.1. Analysis Results of Rainfall, Mechanization, and Agricultural Carbon Emissions

The optimal lag order for agricultural mechanization, rainfall, and agricultural carbon emissions should be determined before VAR systems are used to analyze their pulse effect and variance decomposition as endogenous variables. In this paper, LR test statistic (LR), final prediction error (FPE), Akaike information criterion (AIC), Schwarz information criterion (SIC), and Hannan–Quinn information criterion (HQ) were the five methods for comprehensive judgment. As shown in Table 14, it can be seen that lag order 2 was selected as the optimal lag term. According to this sequence, Figure 2 was produced. It can be seen from Figure 2 that each root was within the unit circle, so this VAR model had the conditions for variance decomposition analysis and impulse response analysis.

The impulse response function can visually describe the VAR model of a standard deviation of the random disturbance impact on other variables’ trajectories and the effect of the influence of current and future values. Therefore, this paper further analyzed the impact of the changes in ACE, agricultural mechanization, and precipitation on the other two variables through an impulse response function diagram. The response time set in this paper was 15 years. In Figure 3, longitudinal coordinates indicate the degree of response of endogenous variables to an impact, the abscissa indicates the lag length of the impact, and dotted lines on either side of the solid lines indicate the range of possible impulse response. The results in Figure 3 are as follows:(1)The impulse response to the impacts of ACE was positive and large, attenuated slightly in the first phase and then was stabilized and sustained. The impulse response of ACE to rainfall and agricultural mechanization was positive in the initial stage, and it turned negative in the fourth stage and converged to 0 in the long run. This indicates that the increase in both will promote the increase in ACE in a short amount of time.(2)The impulse response of agricultural mechanization to ACE was positive in the initial stage, and then tended towards zero after reaching the maximum in the second stage. The impulse response of agricultural mechanization to rainfall was negative at the beginning, turned positive after several periods, and then approached zero, indicating that the increase in ACE and the decrease in rainfall will promote agricultural mechanization to some extent.(3)The initial impulse response of rainfall to agricultural mechanization and ACE was weak, but in the long run, rainfall had a negative response to ACE. This suggests that rainfall is almost unaffected by agricultural mechanization and that ACE has a lagged impact on rainfall.

On the basis of the above analysis, the importance of each structural impact on endogenous variables can be assessed by variance decomposition in VAR models. The calculation results of variance decomposition of the VAR model of ACE, rainfall, and agricultural mechanization are shown in Table 15.

Agricultural carbon emissions are mainly affected by self-impact, followed by agricultural mechanization. With a lag of 15 cycles, the variance contribution rates of rainfall and mechanization were 0.44% and 1.88%, respectively. Agricultural mechanization was mainly affected by its impact and gradually weakened. The influence of rainfall gradually became stronger, indicating that the influence of rainfall on it was a long-term process. Rainfall was mainly affected by its impact, followed by agricultural carbon emissions, but not obviously.

#### 4.7.2. Analysis Results of Sunlight, Mechanization, and Agricultural Carbon Emissions

The optimal lag order for agricultural mechanization, sunlight, and agricultural carbon emissions should be determined before VAR systems can be used to analyze their pulse effect and variance decomposition as endogenous variables. The same five methods used in Section 4.7.1 were used here. As shown in Table 16, it can be seen that the third lag order was selected as the optimal lag term. It can be seen from Figure 4 that each root was within the unit circle, and therefore this VAR model had the conditions for variance decomposition analysis and impulse response analysis.

The assumptions in Figure 5 are similar to those in Figure 3 in Section 4.7.1 and are not detailed here. The results in Figure 4 are as follows:

The impulse response of ACE to its impact was positive and large, and the impulse response of ACE to agricultural mechanization was consistent with the analysis in Section 4.7.2. However, the pulsing response of ACE to sunlight was negative at first, positive over several periods, and then tended towards zero. This was consistent with the results of the FMOLS and DOLS tests, indicating that sunlight can reduce ACE.

The impulse response of sunlight to ACE was always negative, which was strong in the first few periods, and then decreased. These results indicate that ACE would have a certain weakening effect on sunlight. The impulse response of sunlight duration to agricultural mechanization was positive at the beginning, turned negative after several periods, and then approached zero, indicating that sunlight duration has a promoting effect on agricultural mechanization.

Table 17 shows the calculation results of variance decomposition of the VAR model for agricultural carbon emission, sunlight, and agricultural mechanization.

ACE was mainly affected by self-impact, followed by agricultural mechanization. This was highly consistent with the results in Section 4.7.2. Agricultural mechanization was mainly affected by its impact and gradually weakened. Then, it was influenced by carbon emissions from agriculture, but the impact was not significant. Sunlight was mainly affected by its impact, followed by ACE. In phase 15, the contribution of ACE variance reached 3.26%.

## 5. Discussion

We use cross-sectional correlation tests to verify the correlation between variables. In this article, we tested the stability of a unit root of panel data using the IPS test, ADF test, PP test, and LLC test. The results show that the variable after the first order difference was stable, indicating that each variable was an integrated sequence of the same order and can be used in the PVAR model. In addition, we validated the long-term cointegration relationship between variables using the Kao test. The results show that there was a long-term integration relationship between these two groups of variables.

Then, ARDL- and VAR-based impulse response methods were used to conduct an empirical study on the relationship between variables. The results show that the impulse response function more intuitively reflected the dynamic influence between the variables studied in this paper. Results from FOMLS and DOLS tests confirmed the robustness of long-term results. In addition, the Granger causality test was used to study the causal relationship between variables, and the findings can be summarized as follows: First, there was bidirectional causality between rainfall, sunlight, agricultural mechanization, and agricultural carbon emissions. Rainfall and sunlight were Grange causes of agricultural mechanization. Second, agricultural mechanization and rainfall both increased carbon emissions, while sunlight reduced them. In the short term, rainfall reduced the level of agricultural mechanization, and sunlight increased it.

We can speculate the reasons for the interaction between variables by focusing on the empirical results. The results of this paper show that agricultural mechanization will promote agricultural carbon emissions, which is similar to the results of Liu [50] and Feng [51]. The consumption of fossil fuels brought about by agricultural mechanization will undoubtedly increase agricultural carbon emissions. However, agricultural mechanization will increase agricultural productivity, reduce straw burning, and enhance fertilizer utilization to a certain extent. In the long run, technological innovation will reduce mechanized energy consumption and agricultural carbon emissions [52]. At the same time, we found that rainfall has a positive impact on agricultural mechanization and a negative impact on agricultural carbon emissions. This is close to the findings of Barberis [53] and Liu et al. [54]. This phenomenon should be due to the multiple effects of climate change on agricultural production [55]. Climate change has led to unusual droughts in some areas, and low rainfall means increased irrigation equipment [56], which increases energy consumption and further increases agricultural carbon emissions. At the same time, reduced rainfall will also induce abnormal pests in some areas, resulting in increased use of pesticides, leading to more agricultural carbon emissions [57].

## 6. Conclusions and Policy Implications

Global warming has become a major environmental problem facing all countries of the world, one that is linked to the sustainable development of future human life. Moreover, agricultural carbon emissions are significantly connected to climate change. Therefore, we used panel data from 30 provinces (cities) of China between 2000 and 2019, constructing an empirical research framework on the impact of climate change and agricultural mechanization on agricultural carbon emissions, with empirical results. The results can be summarized as follows. Firstly, there was a two-way causal relationship between rainfall, sunlight, agricultural mechanization, and agricultural carbon emissions. Rainfall and sunshine were the main reasons for agricultural mechanization. Second, agricultural mechanization and rainfall will increase carbon emissions, while sunlight will reduce carbon emissions. In the short term, rainfall reduces agricultural mechanization, while sunlight increases it.

On the basis of our findings, we draw the following policy implications: First, with improving resource utilization efficiency and eco-environmental protection at the core, we should increase financial input, promote the renewal and iteration of agricultural machinery equipment, improve agricultural production efficiency, and encourage research and use of renewable energy. At the same time, a large number of low-carbon agricultural technologies, corresponding equipment, and human resources should be introduced to guide and train relevant universities, research institutions, and social organizations. Secondly, each region should optimize the agricultural industrial structure according to local climatic conditions, adjust measures to local conditions, and find its own positioning. Moreover, the application of renewable energy in agriculture should be explored. At the same time, we should build resource-saving and climate-smart agriculture. Furthermore, we should build resource-efficient and climate-smart agriculture. Third, we should strengthen publicity so that farmers and agricultural subjects will be fully aware of the seriousness of climate change. Only in this way can we deeply understand that agriculture is a double-edged sword in climate change and build awareness of low-carbon economic development.

This study is innovative on the basis of predecessors, but there are some limitations: First of all, this paper only selected two climate variables to study, and climate change on agricultural carbon emissions is not systematic. In addition, the direction of agricultural development in China‘s provinces is different, and the impact of agricultural mechanization on agricultural carbon emissions in different regions is also different. This difference between regions has not been reflected in this article. It is suggested that future research should focus on regionally differentiated and multi-level low-carbon policies and government behaviors.

## Figures and Tables

**Figure 1 ijerph-19-14508-f001:**
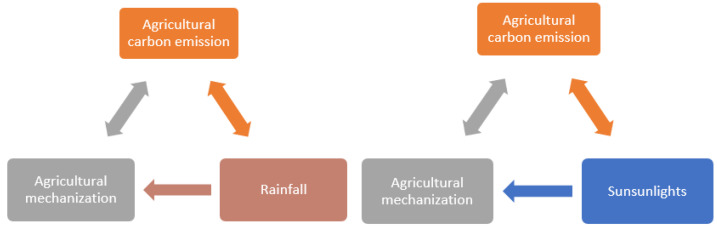
The Granger causality diagram. A pointing to B indicates that A is the Granger cause of B.

**Figure 2 ijerph-19-14508-f002:**
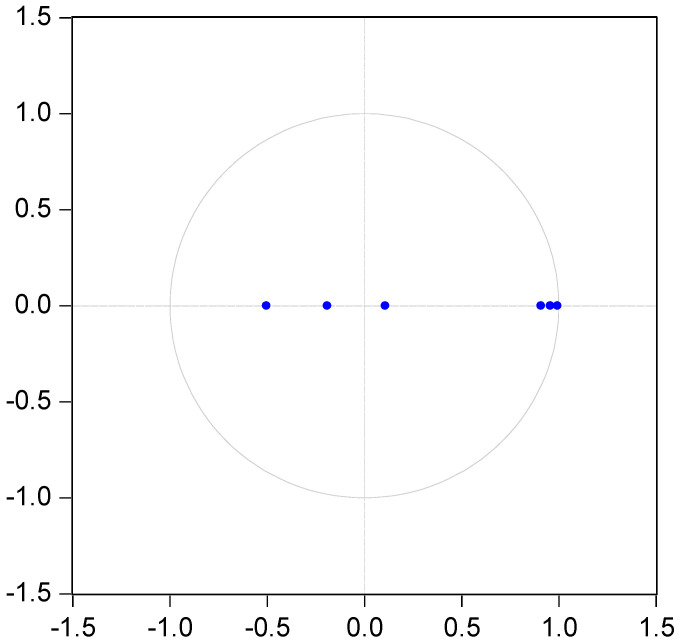
Inverse Roots of PVAR characteristic polynomial.

**Figure 3 ijerph-19-14508-f003:**
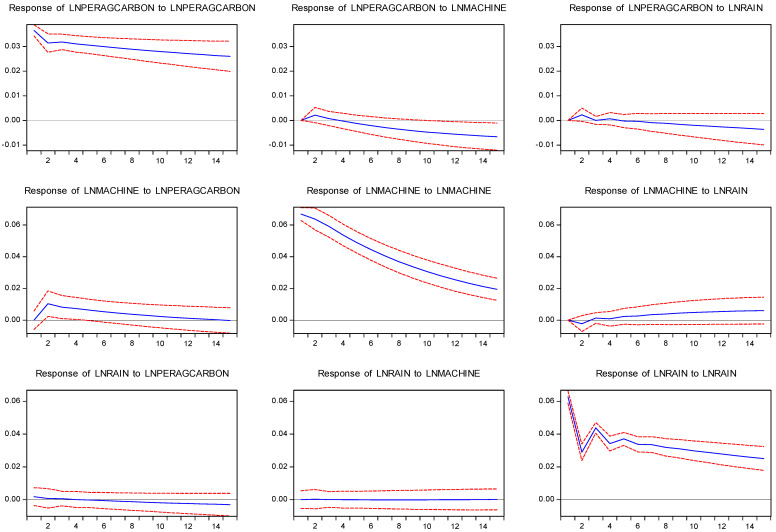
Impulse response of LNRAIN, LNPERAGCARBON, and LNMACHINE from 2004 to 2019 (blue) at a 95% confidence interval (red).

**Figure 4 ijerph-19-14508-f004:**
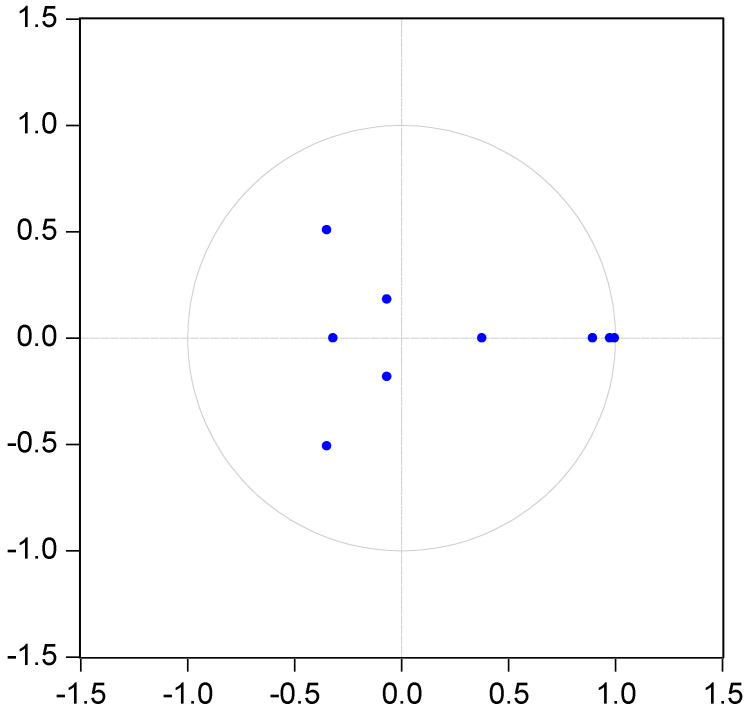
Inverse roots of PVAR characteristic polynomial.

**Figure 5 ijerph-19-14508-f005:**
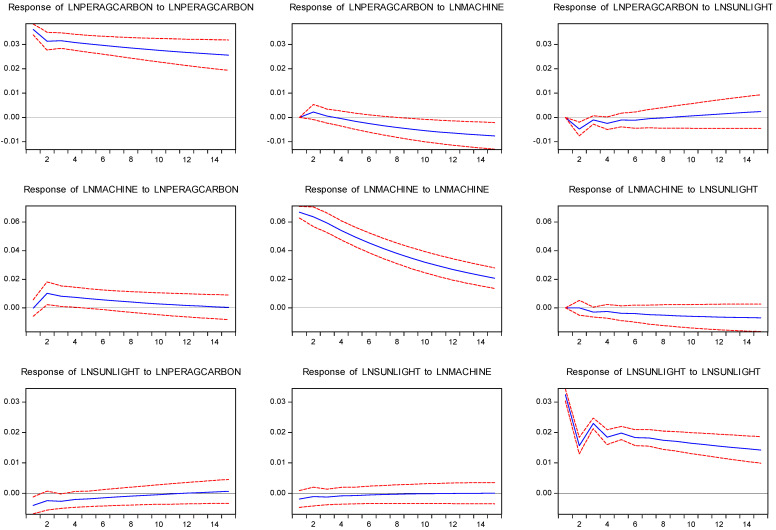
Impulse response of LNSUNLIGHT, LNPERAGCARBON, and LNMACHINE from 2004 to 2019 (blue) at a 95% confidence interval (red).

**Table 1 ijerph-19-14508-t001:** Carbon emission factor reference.

Carbon Emission Source	Carbon Emission Factor	Reference
Fertilizer	0.8955 kg(C)/kg	Oak Ridge National Laboratory [35]
Pesticides	4.9342 kg(C)/kg	Oak Ridge National Laboratory [35]
Agricultural diesel oil	0.5926 kg(C)/kg	Intergovernmental Panel on Climate Change IPCC [36]
Agricultural plastic films	5.18 kg(C)/kg	Institute of Resource, Ecosystem, and Environment of Agriculture, Nanjing Agricultural University
Agricultural power	$0.7921\;t*{\mathrm{MWh}}^{-1}$	China’s Ministry of Ecology and Environment
Agricultural irrigation	266.48 kg(C)/${\mathrm{hm}}^{2}$	Intergovernmental Panel on Climate Change IPCC

**Table 2 ijerph-19-14508-t002:** Definition of variables.

Variables	Definition
Carbon emissions from agricultural production (PERAGCARBON)	Average carbon emissions from agricultural production
Rainfall (RAIN)	The average total rainfall of each province (municipality)
Sunlight intensity (SUNSUNLIGHTS)	The average total sunlight days of each province (city)
Index of mechanization (MACHINE)	Average agricultural mechanization intensity of each province (municipality)

**Table 3 ijerph-19-14508-t003:** Descriptive statistical analysis of main variables.

Variables	Measurements	Mean	Std. Dev	Max	Min
LNSUNLIGHT	600	3.298867	0.110497	3.471238	2.969882
LNMACHINE	600	0.772050	0.277404	1.687066	−0.194059
LNPERAGCARBON	600	−0.565089	0.220080	0.049400	−1.187293
LNRAIN	600	3.924896	0.220364	4.348630	3.302853
PERAGCARBON	600	0.313025	0.190196	1.120470	0.064969
RAIN	600	9488.399	4493.551	22,316.68	2008.411
SUNSUNLIGHTS	600	2052.062	486.8779	2959.632	932.9999
MACHINE	600	7.217916	5.312605	48.64810	0.639648

**Table 4 ijerph-19-14508-t004:** Cross-sectional correlation test results of rainfall, agricultural mechanization, and agricultural carbon emissions.

Test	Statistic	Prob
Breusch–Pagan LM	4262.054	0.0000 ***
Pesaran scaled LM	129.7493	
Bias-corrected scaled LM	128.9160	
Pesaran CD	55.07028	

Note: *** indicates that this column has 1% visibility.

**Table 5 ijerph-19-14508-t005:** Cross-sectional correlation test results of sunlight, agricultural mechanization, and agricultural carbon emissions.

Test	Statistic	Prob
Breusch–Pagan LM	3254.514	0.0000 ***
Pesaran scaled LM	95.59048	
Bias-corrected scaled LM	94.65298	
Pesaran CD	47.37087	

Note: *** indicates that this column has 1% visibility.

**Table 6 ijerph-19-14508-t006:** Kao test results of rainfall, agricultural mechanization, and agricultural carbon emissions.

	Null Hypothesis	*t*-Statistics	Probability
ADF	No co-integration	−1.520172	0.0642

**Table 7 ijerph-19-14508-t007:** Kao test results of sunlight, agricultural mechanization, and agricultural carbon emissions.

	Null Hypothesis	*t*-Statistics	Probability
ADF	No co-integration	−1.659445	0.0485

**Table 8 ijerph-19-14508-t008:** Benchmark results for rainfall and agricultural mechanization.

Variables	Coefficient	SE	*t*-Statistic	Prob
DOLS				
LNMACHINE	0.247467	0.036966	6.694494	0.0000 ***
LNRAIN	0.887540	0.210717	4.212006	0.0000 ***
FMOLS				
LNMACHINE	0.306027	0.031506	9.713155	0.0000 ***
LNRAIN	0.396832	0.096080	4.130199	0.0000 ***

Note: *** indicates that this column has 1% visibility.

**Table 9 ijerph-19-14508-t009:** Benchmark results for sunlight and agricultural mechanization.

Variables	Coefficient	SE	*t*-Statistic	Prob
DOLS				
LNMACHINE	0.314045	0.031825	9.867907	0.0000 ***
LNSUNLIGHT	−0.605700	0.200574	−3.019834	0.0026
FMOLS				
LNMACHINE	0.241701	0.040021	6.039390	0.0000 ***
LNSUNLIGHT	−1.557195	0.543399	−2.865656	0.0045

Note: *** indicates that this column has 1% visibility.

**Table 10 ijerph-19-14508-t010:** ARDL results of agricultural mechanization and rainfall.

Variable	Coefficient	Std. Error	*t*-Statistic	Prob.
	Long run equation			
LNMACHINE	0.417591	0.015647	26.68779	0.0000
LNRAIN	0.125052	0.050689	2.467027	0.0141
	Short run equation			
D(LNMACHINE)	−0.110223	0.113844	−0.968197	0.3337
D(LNMACHINE(-1))	0.081768	0.061751	1.324157	0.1864
D(LNMACHINE(-2))	0.058224	0.079353	0.733734	0.4636
D(LNRAIN)	−0.113511	0.053012	−2.141213	0.0330
D(LNRAIN(-1))	−0.035060	0.060387	−0.580584	0.5619
D(LNRAIN(-2))	−0.052880	0.047000	−1.125106	0.2614
C	−0.380977	0.143890	−2.647701	0.0085

**Table 11 ijerph-19-14508-t011:** ARDL results of agricultural mechanization and sunlight.

Variable	Coefficient	Std. Error	*t*-Statistic	Prob.
	Long run equation			
LNMACHINE	0.486935	0.023574	20.65570	0.0000
LNSUNLIGHT	−0.410620	0.156628	−2.621625	0.0090
	Short run equation			
COINTEQ01	−0.257725	0.051858	−4.969813	0.0000
D(LNMACHINE)	−0.049302	0.049635	−0.993309	0.3211
D(LNSUNLIGHT)	−0.057476	0.064412	−0.892309	0.3727
C	0.112503	0.024569	4.579091	0.0000

**Table 12 ijerph-19-14508-t012:** Granger causality test results of rainfall, mechanization, and agricultural carbon emissions.

Null Hypothesis	F-Statistic	Prob.
LNMACHINE does not cause LNPERAGCARBON via Granger test	3.54771	0.0037
LNPERAGCARBON does not cause LNMACHINE via Granger test	3.53892	0.0038
LNRAIN does not cause LNPERAGCARBON via Granger test	2.28559	0.0454
LNPERAGCARBON does not cause LNRAIN via Granger test	6.73996	0.0000
LNRAIN does not cause LNMACHINE via Granger test	2.10853	0.0634
LNMACHINE does not cause LNRAIN via Granger test	0.45362	0.8107

**Table 13 ijerph-19-14508-t013:** Granger causality test results of sunlight, mechanization, and agricultural carbon emissions.

Null Hypothesis:	F-Statistic	Prob.
LNMACHINE does not cause LNPERAGCARBON via Granger test	3.54771	0.0037
LNPERAGCARBON does not cause LNMACHINE via Granger test	3.53892	0.0038
LNSUNLIGHT does not cause LNPERAGCARBON via Granger test	3.44080	0.0046
LNPERAGCARBON does not cause LNSUNLIGHT via Granger test	2.24957	0.0486
LNSUNLIGHT does not cause LNMACHINE via Granger test	1.72609	0.1272
LNMACHINE does not cause LNSUNLIGHT via Granger test	0.42075	0.8343

**Table 14 ijerph-19-14508-t014:** Optimal lag period selection result.

Lag	LogL	LR	FPE	AIC	SC	HQ
0	214.9547	NA	$4.89\times 10^{-5}$	−1.413032	−1.375994	−1.398209
1	1644.218	2820.414	$3.77\times 10^{-9}$	−10.88146	−10.73330	−10.82217
2	1754.703	215.8126	$1.92\times 10^{-9}$	−11.55802	−11.29875 ***	−11.45426
3	1772.115	33.66480	$1.81\times 10^{-9}$	−11.61410	−11.24372	−11.46588
4	1783.863	22.47780	$1.78\times 10^{-9}$	−11.63242	−11.15093	−11.43973
5	1799.004	28.66656	$1.71\times 10^{-9}$	−11.67336	−11.08076	−11.43620

Note: *** indicates that this column has 10% visibility.

**Table 15 ijerph-19-14508-t015:** The variance decomposition results.

Variance Decomposition of LNPERAGCARBON:				
Period	S.E.	LNPERAGCARBON	LNMACHINE	LNRAIN
1	0.036	100.000	0.000	0.000
2	0.048	99.574	0.201	0.225
3	0.058	99.688	0.155	0.157
4	0.066	99.746	0.122	0.132
5	0.072	99.760	0.130	0.110
6	0.078	99.721	0.183	0.096
7	0.084	99.626	0.278	0.096
8	0.089	99.488	0.408	0.104
9	0.093	99.307	0.569	0.124
10	0.097	99.093	0.754	0.153
11	0.101	98.850	0.958	0.193
12	0.105	98.583	1.176	0.242
13	0.109	98.295	1.404	0.301
14	0.112	97.993	1.638	0.369
15	0.115	97.677	1.877	0.446
**Variance Decomposition of LNMACHINE:**				
**Period**	**S.E.**	**LNPERAGCARBON**	**LNMACHINE**	**LNRAIN**
1	0.067	0.000	100.000	0.000
2	0.093	1.245	98.701	0.054
3	0.110	1.434	98.513	0.052
4	0.123	1.515	98.438	0.047
5	0.133	1.532	98.396	0.073
6	0.140	1.521	98.376	0.103
7	0.146	1.497	98.350	0.153
8	0.150	1.466	98.321	0.212
9	0.154	1.433	98.282	0.285
10	0.157	1.400	98.232	0.368
11	0.160	1.367	98.172	0.461
12	0.162	1.338	98.100	0.562
13	0.164	1.311	98.018	0.671
14	0.165	1.288	97.926	0.786
15	0.167	1.269	97.825	0.906
**Variance Decomposition of LNRAIN:**				
**Period**	**S.E.**	**LNPERAGCARBON**	**LNMACHINE**	**LNRAIN**
1	0.063	0.086	0.000	99.914
2	0.069	0.083	0.001	99.916
3	0.082	0.065	0.001	99.934
4	0.089	0.056	0.001	99.944
5	0.096	0.048	0.001	99.951
6	0.102	0.046	0.001	99.953
7	0.107	0.049	0.001	99.950
8	0.112	0.057	0.001	99.942
9	0.116	0.071	0.001	99.928
10	0.120	0.090	0.001	99.910
11	0.123	0.114	0.001	99.886
12	0.126	0.143	0.001	99.857
13	0.129	0.176	0.001	99.823
14	0.132	0.215	0.001	99.785
15	0.134	0.258	0.001	99.742

**Table 16 ijerph-19-14508-t016:** Optimal lag period selection result.

Lag	LogL	LR	FPE	AIC	SC	HQ
0	398.3909	NA	$1.44\times 10^{-5}$	−2.635939	−2.598901	−2.621117
1	1840.989	2846.726	$1.02\times 10^{-9}$	12.19326	−12.04511	−12.13397
2	1939.346	192.1246	$5.60\times 10^{-10}$	−12.78897	−12.52971	−12.68521
3	1967.990	55.37755	$4.92\times 10^{-10}$	−12.78897	12.54955 ***	−12.77170
4	1982.863	28.45888	$4.73\times 10^{-10}$	−12.78897	−12.47760	−12.76640
5	2004.334	40.65023	$4.35\times 10^{-10}$	−12.78897	−12.44962	−12.80506
6	2017.730	25.09534	$4.23\times 10^{-10}$	−12.78897	−12.36781	−12.78990

Note: *** indicates that this column has 10% visibility.

**Table 17 ijerph-19-14508-t017:** The variance decomposition results.

Variance Decomposition of LNPERAGCARBON:				
Period	S.E.	LNPERAGCARBON	LNMACHINE	LNSUNLIGHT
1	0.037	100.000	0.000	0.000
2	0.049	99.486	0.095	0.419
3	0.060	99.386	0.170	0.444
4	0.069	99.521	0.139	0.340
5	0.076	99.600	0.123	0.277
6	0.083	99.576	0.163	0.260
7	0.089	99.495	0.268	0.236
8	0.094	99.332	0.441	0.227
9	0.098	99.100	0.668	0.232
10	0.103	98.821	0.941	0.238
11	0.107	98.495	1.252	0.253
12	0.110	98.134	1.591	0.276
13	0.114	97.748	1.949	0.303
14	0.117	97.341	2.321	0.337
15	0.120	96.921	2.701	0.378
**Variance Decomposition of LNMACHINE:**				
**Period**	**S.E.**	**LNPERAGCARBON**	**LNMACHINE**	**LNSUNLIGHT**
1	0.068	0.028	99.972	0.000
2	0.094	1.266	98.701	0.033
3	0.112	2.213	97.758	0.029
4	0.125	2.653	97.323	0.024
5	0.134	2.889	97.089	0.022
6	0.142	2.995	96.977	0.028
7	0.148	3.030	96.927	0.043
8	0.152	3.024	96.912	0.064
9	0.156	2.996	96.911	0.094
10	0.158	2.955	96.913	0.132
11	0.161	2.910	96.912	0.178
12	0.162	2.864	96.905	0.232
13	0.164	2.820	96.887	0.292
14	0.165	2.781	96.859	0.360
15	0.166	2.748	96.818	0.434
**Variance Decomposition of LNSUNLIGHT:**				
**Period**	**S.E.**	**LNPERAGCARBON**	**LNMACHINE**	**LNSUNLIGHT**
1	0.031	1.033	0.182	98.784
2	0.033	2.620	0.179	97.200
3	0.035	3.388	0.331	96.281
4	0.039	3.399	0.316	96.285
5	0.041	3.788	0.300	95.912
6	0.043	3.884	0.286	95.831
7	0.046	3.861	0.260	95.880
8	0.048	3.871	0.241	95.889
9	0.049	3.813	0.223	95.964
10	0.051	3.733	0.210	96.057
11	0.053	3.652	0.201	96.147
12	0.054	3.557	0.194	96.249
13	0.055	3.459	0.190	96.351
14	0.057	3.361	0.188	96.451
15	0.058	3.263	0.188	96.549

## Data Availability

Data supporting the conclusions of this article are included within the article. The datasets presented in this study are available on request from the corresponding author.

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
