# Peer review of "Dynamic Linkages among Climate Change, Mechanization and Agricultural Carbon Emissions in Rural China"

_ijerph, 2022, doi:10.3390/ijerph192114508_

Round 1

Reviewer 1 Report

see attached

Author Response

Response to Reviewer 1 Comments

First of all, we would like to thank Reviewer 1 for reading our article and for your valuable comments. Below is our response to all comments made.

Abstract: Add one/two clear sentences of the policy implications at the end of the abstract.

Response 1 : We accepted the reviewer 's recommendation to include our specific policy at the end of the summary. Here 's what we added.

Finally, this paper investigates the green path suitable for the low-carbon development of Chinese agriculture ,and believes that the government should formulate low-carbon agricultural policies by region and actively promote the upgrading of agricultural machinery.

Introduction: I would suggest removal of first paragraph (line 29-50) and its replacement with direct introduction of the agriculture led carbon emissions and their possible consequences with the environment. The consequences of global warming are well known, therefore, direct introduction of the problem in the introduction section is suggested.

Response 2 : We accepted the reviewer 's suggestion and revised the first paragraph of the introduction, retaining only a little global warming content as an introduction, increasing the description of China 's agricultural carbon emissions and its severity. The following is our revised first paragraph.

Climate change has become a common problem faced by all countries. During the 20th century, the average global temperature rose about 1 degree Celsius [1], which led to sea level rise [2], species extinction [3], and frequent climate extremes [4]. China's carbon dioxide emissions hit 11.9 billion tons in 2021, making up 33 percent of the world’s total and far outpacing those of other nations and regions, according to the International Energy Agency (IEA). As a consequence, controlling China's carbon emissions is vital for mitigating climate change. The Chinese government has also shown its determination to tackle global warming. China has pledged to peak its carbon emissions by 2030 and become carbon neutral by 2060, which is referred to as the "dual carbon" target. In China, about 20 percent of carbon emissions are attributable to agricultural production and land use [5], and agriculture-related carbon emissions make a contribution of about a quarter of the globe’s total [6]. Due to the extensiveness and universality of China 's agricultural production activities and the dispersion of agricultural production entities, coupled with the wide range, randomness, concealment, difficulty in monitoring and quantification of agricultural carbon emissions, China 's agricultural carbon emissions are difficult to control. Meanwhile, China 's agricultural production is large and growing rapidly. If there is no corresponding reduction measures, agricultural greenhouse gas emissions will continue to rise rapidly. Therefore, in order to achieve the " double carbon " goal, China must find a way to reduce agricultural carbon emissions as soon as possible, and realize the transformation of agricultural production to green, low-carbon and sustainable development.

Literature Review: This section is written in a good way. However, some most relevant

studies are not part of the literature. Their inclusion may enrich and update this section. Among others, I would suggest few studies that may be considered.

Response 3 : Thank you very much for the reviewer 's endorsement of this section. We have listened to the reviewer 's comments. In this section, we have included literature on the relationship between mechanization and climate change

Materials and Methods: Rename section 1.1. Currently it is named as “subsection” that does not make sense. Agriculture land use is an important variable that is linked with carbon emissions. Add few sentences for not choosing this variable for the model. Also explain why sunlight is preferred over temperature. In Table 2, there is need to explain why some variables are in log form and the other in general form. Despite depth analysis and well execution of the econometric methodology, results obtained are mere listed and not interpreted. I mean discussion of the result is missing. I would suggest two or three paragraphs at the end of “material and method” Section that link the major findings

with previous empirical literature (authors can also add a separate section “discussion” if they intend to, however it is optional).

Response 3 : We accepted the reviewer 's opinion and renamed the first section of this part as Calculation of Agricultural Carbon Emission At the same time in the text to increase the illumination time as a variable compared with the advantages of temperature, and land use we think its correlation with climate change is not strong, so no choice. We also explain the reasons for taking the logarithm at the top of Table 3, and we add a section to discuss the empirical results ( this section was originally in the last section )

The discussion section is as follows : We use cross-sectional correlation test to verify the correlation between variables.. In this article, we tested the stability of a unit root of panel data using the IPS test, ADF test, PP test, and LLC test. The results show that the variable after the first order dif-ference is stable, indicating that each variable is an integrated sequence of the same order and can be used in the PVAR model. In addition, we validated the long-term cointegration relationship between variables using the Kao test. The results show that there is a long-term-integration relationship between these two groups of variables.

Then ARDL and VAR-based impulse response methods are used to conduct an empirical study on the relationship between variables. The results show that the im-pulse response function more intuitively reflects the dynamic influence between the variables studied in this paper. Results from FOMLS and DOLS tests confirm the ro-bustness of long-term results. In addition, the Granger causality test is used to study the causal relationship between variables, and the findings can be summarized as fol-lows. First, there is bidirectional causality between rainfall, sunlights, agricultural mechanization, and agricultural carbon emissions. Rainfall and sunlights are Grange causes of agricultural mechanization. Second, agricultural mechanization and rainfall both increase carbon emissions, while sunlights reduces them. In the short term, rain-fall reduces the level of agricultural mechanization, and sunlights increases it.

We can speculate the reasons for the interaction between variables by focusing on the empirical results. The results of this paper show that agricultural mechanization will promote agricultural carbon emissions, which is similar to the results of Liu [50] and Feng [51]. The consumption of fossil fuels brought by agricultural mechanization will undoubtedly increase agricultural carbon emissions. However, agricultural mechanization will increase agricultural productivity and reduce straw burning and enhance fertilizer utilization to a certain extent. In the long run, technological innova-tion will reduce mechanized energy consumption and agricultural carbon emissions. [52]. At the same time, we found that rainfall has a positive impact on agricultural ag-ricultural mechanization and a negative impact on agricultural carbon emissions. This is close to the findings of Barberis [53] and Liu [54] et al. This phenomenon should be due to the multiple effects of climate change on agricultural production [55]. Climate change has led to unusual droughts in some areas, and low rainfall means increased irrigation equipment [56], which increases energy consumption and further increases agricultural carbon emissions. At the same time, reduced rainfall will also induce ab-normal pests in some areas, resulting in increased use of pesticides, resulting in more agricultural carbon emissions [57].

Conclusions and policy implications: Instead of giving background of the issue, just start from your main findings and then draw policy implications based on these findings. I would suggest that this section may be reduced to 1/3 of the current.

Response 4 : We thank the reviewers for their comments. There was originally a discussion of the empirical results in the last part, and now we put them separately in another part. Now the last part has been greatly shortened

Reviewer 2 Report

Dear Editor/Authors

I studied this manuscript with interest and in my view, it is a good article that attempts to analyze the dynamic connection between climate change, mechanization, and agricultural carbon emissions. I have a few major and minor comments/suggestions. I suggest the acceptance of this manuscript after minor revision. The detailed comments are as follows:

Minor comments/suggestions:

1        To facilitate the reader, the number should be enlarged for the impulse response diagram.  There is a spelling mistake in Figure 1, and please increase the font size in the tables.

2        The limitations of the study should be pointed out in the paper for the readers.

3        I suggest replacing the word “mechanization” with “agricultural mechanization” throughout the manuscript.

4        The reference format and style must be according to the journal.

5        The symbols, format, font, etc. of some mathematical formulas need correction.

6        Pay more attention to the concluding section of the study for policy implications.

7        The English quality is better however, I suggest reviewing from a native speaker.

Major comments/suggestions:

8        The details about all the tests (diagnostic tests, co-integration, etc.) should be removed because it looks more like a book chapter than a research article.

9        The paper has listed a list of econometric procedures, no baseline test was found, such as the Hausman specification test. Please, consider if required for the quality of the manuscript.

10    It is suggested to use Hausman fixed or random effect. In addition, year fixed effect and province fixed effect.

11    No test was employed for dealing with endogeneity. The readers would benefit if clarified.

12    Line 428, do you mean tables 13,14 or tables 12 &13 for granger causality?

13    Since the study used Panel ARDL overall, it would be better if each province were listed in a table.

14    There are 17 tables, I would suggest removing or transferring some tables to the supplementary materials.   I suggest reducing the decimal points to two or three. For example, 0.00 or 0.000 is enough. If you write 0.000000, it will cause confusion and increase the table size.

15    The paper is too lengthy, and the formulas are confusing, if possible, please concise the paper and move some parts to supplementary data. 

Author Response

Response to Reviewer 2 Comments

First of all, we would like to thank Reviewer 2 for reading our article and for your valuable comments. Below is our response to all comments made.

1.To facilitate the reader, the number should be enlarged for the impulse response diagram.  There is a spelling mistake in Figure 1, and please increase the font size in the tables.

Response 1 : Thanks to the reviewer 's reminder, we have fixed the spelling errors in Figure 1 and increased the fonts in the impulse response plots and tables.

2.The limitations of the study should be pointed out in the paper for the readers.

Response 2 : We accepted the reviewer ' s opinion. At the end of the article, we add the shortcomings of this study and put forward some suggestions for future research directions. The supplementary contents are as follows.

This study is innovative on the basis of predecessors, but there are some limita-tions : First of all, this paper only selected two climate variables to study, climate change on agricultural carbon emissions is not systematic. In addition, the direction of agricultural development in China 's provinces is different, and the impact of agricul-tural mechanization on agricultural carbon emissions in different regions is also dif-ferent. This difference between regions has not been reflected in this article. It is sug-gested that future research should focus on regionally differentiated and multi-level low-carbon policies and government behaviors.

3.I suggest replacing the word “mechanization” with “agricultural mechanization” throughout the manuscript.

Response 3 : We have changed most of the mechanization in the text to agricultural mechanization.

4.The reference format and style must be according to the journal.

Response 4 : We have adjusted the document format to meet the journal requirements.

5.The symbols, format, font, etc. of some mathematical formulas need correction.

Response 5 : Thanks for the reviewer 's reminder. We checked the fonts and symbols of the full text and modified the italics in some formulas.

6.Pay more attention to the concluding section of the study for policy implications.

Response 6 : We accepted the reviewer ' s opinion. We have detailed some of the policy recommendations made in this paper.

7.The English quality is better however, I suggest reviewing from a native speaker.

Response 7 : Thanks to the reviewers for their comments, we and English native speakers have embellished the language of the full text

8.The details about all the tests (diagnostic tests, co-integration, etc.) should be removed because it looks more like a book chapter than a research article.

Response 8 : Thanks to the reviewer 's advice, we moved the detailed process of cross-sectional testing methods and panel unit root testing to the appendix, making the article less lengthy

9.The paper has listed a list of econometric procedures, no baseline test was found, such as the Hausman specification test. Please, consider if required for the quality of the manuscript.

Response 9 : This study does not do panel random effects and fixed effects. What we do is panel VAR.According to the principle of panel VAR, we need to test whether the variables have unit roots, whether the variables are single and integral, and whether there is a cointegration relationship between the variables. No need to use the hausman test to see if the model is a fixed effects model or a random effects model

10.t is suggested to use Hausman fixed or random effect. In addition, year fixed effect and province fixed effect.

Response 10 : As the response 9, according to the principle of panel VAR, we need to test whether there is a unit root of the variables, whether the variables are single, and whether there is a cointegration relationship between the variables. No need to use the hausman test to see if the model is a fixed effects model or a random effects model

11.No test was employed for dealing with endogeneity. The readers would benefit if clarified.

Response 11 : According to the research paradigm of PVAR and referring to the previous literature, the discussion of endogenous problems is not involved.

12.Line 428, do you mean tables 13,14 or tables 12 &13 for granger causality?

Response 12 : I am sorry that we made a mistake in adjusting the serial number of the table. The Granger causality between Table 12 and Table 13 is shown here.

13.Since the study used Panel ARDL overall, it would be better if each province were listed in a table.

Response 13 : Due to the length of the article, this paper pays more attention to the generality of the research conclusions, and does not have much energy to explore the heterogeneity of long-term and short-term relationships in various provinces. Future research will explore this issue.

14.There are 17 tables, I would suggest removing or transferring some tables to the supplementary materials.   I suggest reducing the decimal points to two or three. For example, 0.00 or 0.000 is enough. If you write 0.000000, it will cause confusion and increase the table size.

Response 14 : We move the panel unit root test results to Appendix C in accordance with the opinions of reviewers. At the same time we will variance decomposition results are retained three decimal places to ensure that the results are accurate and beautiful table. The following is one of the tables that we modified.

Variance Decomposition of LNPERAGCARBON:

Period

S.E.

LNPERAGCARBON

LNMACHINE

LNSUNLIGHT

1

0.037

100.000

0.000

0.000

2

0.049

99.486

0.095

0.419

3

0.060

99.386

0.170

0.444

4

0.069

99.521

0.139

0.340

5

0.076

99.600

0.123

0.277

6

0.083

99.576

0.163

0.260

7

0.089

99.495

0.268

0.236

8

0.094

99.332

0.441

0.227

9

0.098

99.100

0.668

0.232

10

0.103

98.821

0.941

0.238

11

0.107

98.495

1.252

0.253

12

0.110

98.134

1.591

0.276

13

0.114

97.748

1.949

0.303

14

0.117

97.341

2.321

0.337

15

0.120

96.921

2.701

0.378

Variance Decomposition of LNMACHINE:

Period

S.E.

LNPERAGCARBON

LNMACHINE

LNSUNLIGHT

1

0.068

0.028

99.972

0.000

2

0.094

1.266

98.701

0.033

3

0.112

2.213

97.758

0.029

4

0.125

2.653

97.323

0.024

5

0.134

2.889

97.089

0.022

6

0.142

2.995

96.977

0.028

7

0.148

3.030

96.927

0.043

8

0.152

3.024

96.912

0.064

9

0.156

2.996

96.911

0.094

10

0.158

2.955

96.913

0.132

11

0.161

2.910

96.912

0.178

12

0.162

2.864

96.905

0.232

13

0.164

2.820

96.887

0.292

14

0.165

2.781

96.859

0.360

15

0.166

2.748

96.818

0.434

Variance Decomposition of LNSUNLIGHT::

Period

S.E.

LNPERAGCARBON

LNMACHINE

LNSUNLIGHT

1

0.031

1.033

0.182

98.784

2

0.033

2.620

0.179

97.200

3

0.035

3.388

0.331

96.281

4

0.039

3.399

0.316

96.285

5

0.041

3.788

0.300

95.912

6

0.043

3.884

0.286

95.831

7

0.046

3.861

0.260

95.880

8

0.048

3.871

0.241

95.889

9

0.049

3.813

0.223

95.964

10

0.051

3.733

0.210

96.057

11

0.053

3.652

0.201

96.147

12

0.054

3.557

0.194

96.249

13

0.055

3.459

0.190

96.351

14

0.057

3.361

0.188

96.451

15

0.058

3.263

0.188

96.549

15.The paper is too lengthy, and the formulas are confusing, if possible, please concise the paper and move some parts to supplementary data. 

Response 15 : We listened to the reviewers. We move the calculation formula of cross-section test method to Appendix A, and the five methods of panel unit root test to Appendix B. The length of the paper is effectively reduced.

Thank you for your attention and time, look forward to getting your reply as soon as possible

Reviewer 3 Report

Please, see Comments file attached.

Author Response

Response to Reviewer 3 Comments

First of all, we would like to thank Reviewer 1 for reading our article and for your valuable comments. At the same time because of the change, the article text order has changed, change the specific location we will point out in the reply. Below is our response to all comments made.

Line 32

insert a space between rise and [2]

insert a space between extinction and [3]

insert a space between extreme and [4]

Response 1 : We have modified the introduction, and now there is no such line.

Line 51 introduces the acronym GHG without previous explanation of its meaning

Response 2 : Thanks to the reviewer ' s reminder, we have added the meaning to the GHG ( this change is in line 52 ).

Line 53 insert a space between Organization and [7]

Response 4 : Thanks to the reviewer ' s reminder, we have added spaces.

Line 55 insert a space between general and [8]

Response 5 : Thanks to the reviewer ' s reminder, we have added spaces.

Line 58 insert a space between weather and [9]

Response 6 : Thanks to the reviewer ' s reminder, we have added spaces.

Line 59 insert a space between rise and [10]

Response 7 : Thanks to the reviewer ' s reminder, we have added spaces.

Line 65 insert a space between productivity and [13]

Response 8 : Thanks to the reviewer ' s reminder, we have added spaces.

Line 70 insert a space between production and [15]

Response 9 : Thanks to the reviewer ' s reminder, we have added spaces.

Line 76 introduce the acronyms ARDF and PVAR without first explaining their

Meaning

Response 10 : We accepted the reviewer 's suggestion and added the meaning to the abbreviation ( changes in 79 and 80 lines )

Line 97 insert a space between production and [16]

Response 11 : Thanks to the reviewer ' s reminder, we have added spaces.

Line 100 insert a space between growth and [17]

Response 12 : Thanks to the reviewer ' s reminder, we have added spaces.

Line 102 insert a space between environment and [18]

Response 13 : Thanks to the reviewer ' s reminder, we have added spaces.

Line 107 insert a space between effects and [19]

Response 14 : Thanks to the reviewer ' s reminder, we have added spaces.

Line 109 insert a space between patterns and [20]

Response 15 : Thanks to the reviewer ' s reminder, we have added spaces.

Line 111 insert a space between yields and [21]

Response 16 : Thanks to the reviewer ' s reminder, we have added spaces.

Line 113 insert a space between equipment and [24]

Response 17 : Thanks to the reviewer ' s reminder, we have added spaces.

Line 115 insert a space between mechanization and [25]

Response 18 : Thanks to the reviewer ' s reminder, we have added spaces.

Line 126 insert a space between farmland and [26]

Response 19 : Thanks to the reviewer ' s reminder, we have added spaces.

Line 131 insert a space between atmosphere and [27]

Response 20 : Thanks to the reviewer ' s reminder, we have added spaces.

Line 132 insert a space between conditions and [28]

Response 21 : Thanks to the reviewer ' s reminder, we have added spaces.

Line 135 insert a space between diseases and [23]

Response 22 : Thanks to the reviewer ' s reminder, we have added spaces.

Line 136 insert a space between environment and [29]

Response 23 : Thanks to the reviewer ' s reminder, we have added spaces.

Line 139 insert a space between machinery and [30]

Response 24 : Thanks to the reviewer ' s reminder, we have added spaces.

Line 142 insert a space between emissions and [31]

Response 25 : Thanks to the reviewer ' s reminder, we have added spaces.

Line 145 insert a space between emissions and [32]

Response 26 : Thanks to the reviewer ' s reminder, we have added spaces.

Line 168 insert a space between excretion and [33]

Table 1 insert a space between Laboratory and [34], twice

insert a space between IPCC and [35]

Response 27 : Thanks to the reviewer ' s reminder, we have added spaces.

Line 186 insert a space between sunlights and [36]

Response 28 : Thanks to the reviewer ' s reminder, we have added spaces.

insert a space between agriculture and [37]

Response 29 : Thanks to the reviewer ' s reminder, we have added spaces.

Line 195 fix table 1 to table 2

197 to 205 improve the discussion, referring to the values of table 3 that allow these

conclusions

Table 2 insert a space between Mechanization and (MACHINE)

Response 30 : We have fixed the table marking problem.At the same time, we have increased the discussion of the data in Table 3, and also added spaces in Table 2.

Line 205 insert a period

Response 31 : We have added data periods to the descriptive statistics ( change at 204 lines ).

Line 206 fix table 2 to table 3

Response 32 : We have corrected the table marking error.

Line 211 insert a space between data and [38]

Response 33 : Thanks to the reviewer ' s reminder, we have added spaces.

Line 212 insert a space between correlation and [39]

Response 34 : Thanks to the reviewer ' s reminder, we have added spaces.

Line 213 Replace Breusch-Paganlm with Breusch-Pagan Lagrange Multiplier (LM)

Response 35 : We listened to the reviewer 's opinion and completed the replacement of vocabulary.

Line 214 Replace Breusch-Paganlm with Breusch-Pagan LM

Response 36 : We listened to the reviewer 's opinion and completed the replacement of vocabulary.

Line 215 Replace Pesaran CD with Pesaran Cross-sectional Dependence (CD)

Response 37 : We listened to the reviewer 's opinion and completed the replacement of vocabulary.

Line 216 insert a space between model and [40]

Response 38 : Thanks to the reviewer ' s reminder, we have added spaces.

Line 224 write equation (3) in capital letters, to make it uniform

Response 39 : In order to reduce the length of the text, we moved this part of the formula to Appendix A, and we revised the formula A3 ( the original formula ( 3 ) ).

Line 225 insert a period after samples and before In

Response 40 : This part was moved to Appendix A and we added a period.

Line 231 remove dot between difference and [41]

Response 41 : We have deleted that point, and since we added a new document, it is now [ 42 ].

Line 234 Remove these five methods,

Response 42 : We have moved these five methods to Appendix B.

Line 236 DF presumably refers to the Dicky and Fuller model, but is not referred to

earlier, and should

Response 43 : This part has been moved to Appendix B, and we also explain the meaning of abbreviations. The DF model refers to the augmented Dickey-Fuller test.

Line 237 AR model means autoregressive model? Should mention.

Response 44 : This part has been moved to Appendix B, and we also explain the meaning of abbreviations. AR model refers to Autoregressive Model.

Line 282 insert a space between studies and [42]

Response 45 : Thanks to the reviewer ' s reminder, we have added spaces.

Line 298 insert a space between test and [43]

Response 46 : Thanks to the reviewer ' s reminder, we have added spaces.

Line 311 insert a space between Granger and [44]

Response 47 : Thanks to the reviewer ' s reminder, we have added spaces.

Line 312 insert a space between Hurlin and [45]

Response 48 : Thanks to the reviewer ' s reminder, we have added spaces.

Line 319 fix 3.7 by 3.8

Response 49 : We have corrected the table marking error.

Line 321 insert a space between Pesaran and [46]

Response 50 : Thanks to the reviewer ' s reminder, we have added spaces.

Line 331

introduces the acronyms FMOLS and DOLS without first explaining their

meaning (Full Modified Ordinary Least Square (FMOLS) model and Dynamic

Ordinary Least Squares (DOLS) model)

Response 51 : We listened to the reviewer 's comments and introduced these two abbreviations in the text

Line 333 introduces the acronym OLS without first explaining its meaning Ordinary

Least Squares

Response 52 : We listened to the reviewer 's opinion, we added the meaning of abbreviations.

Line 351 insert a space between Lanne and [49]

Response 53 : Thanks to the reviewer ' s reminder, we have added spaces.

Line 369 fix table 3 by table 4

Response 54 : We have corrected the table marking error.

Line 372 fix table 4 by table 5

Response 55 : We have corrected the table marking error.

Line 384 fix table 5 by table 6

Response 56 : We have corrected the table marking error.

Line 393 fix table 6 by table 7

Response 57 : We have corrected the table marking error.

Line 395 fix table 7 by table 8

Response 58 : We have corrected the table marking error.

Line 404 fix table 8 by table 9

Response 59 : We have corrected the table marking error.

Line 406 fix table 9 by table 10

Response 60 : We have corrected the table marking error.

Line 423 fix table 10 by table 11

Response 61 : We have corrected the table marking error.

Line 424 fix table 11 by table 12

Response 62 : We have corrected the table marking error.

Line 432 fix table 12 by table 13

Response 63 : We have corrected the table marking error.

Line 433 fix table 13 by table 14

Response 64 : We have corrected the table marking error.

Line 452 fix table 14 by table 15

Response 65 : We have corrected the table marking error.

Line 453 increase space between figure title and previous line

Response 66 : Thanks to the reviewer for reminding us that we have added spaces

Line 455 insert period

Response 67 : We have added the study time interval.

Line 457 insert period

Response 68 : We have added the study time interval.

Line 481 fix table 15 by table 16

Response 69 : We have corrected the table marking error.

Line 485 Replace Table 15 with Table 16

Response 70 : We have corrected the table marking error.

Line 500 Replace Table 16 with Table 17

insert period

Figure 4 increase space between figure title and previous line

Response 71 : We have corrected the table marking error and added the study time interval

Line 502 insert period

Response 72 : We supplemented the research period

Line 505 insert period

Response 73 : We supplemented the research period

Line 506 there is no section 4.6.3. Correct please

Response 74 : I 'm sorry to us for the error, we 've fixed this error to 4.7.2

Line 509 there is no section 4.6.2. Correct please

Response 75 : I 'm sorry to us for the error, we 've fixed this error to 4.7.2

Line 518

Replace Table 17 with Table 18

This table is referenced after being presented in the text and should be,

preferably, before

Response 76 : We have corrected the table number error and the table is located before the reference location.

Line 522 there is no section 4.6.2. Correct please

Response 77 : I 'm sorry to us for the error, we 've fixed this error to 4.7.2

Line 552 insert a space between Liu and [50]

Response 79 : Thanks to the reviewer ' s reminder, we have added spaces.

Line 553 insert a space between Feng and [51]

Response 80 : Thanks to the reviewer ' s reminder, we have added spaces.

Line 556 insert a space between emissions and [52]

Response 81 : Thanks to the reviewer ' s reminder, we have added spaces.

Line 558 insert a space between Barberis and [53]

replace Liu[54] et al. with Liu et al. [54].

Response 82 : Thanks to the reviewer ' s reminder, we have added spaces.

Line 560 insert a space between production and [55]

Response 83 : Thanks to the reviewer ' s reminder, we have added spaces.

Thank you for your attention and time, look forward to getting your reply as soon as possible
